# Could Fibrinogen Concentration Be a Useful Indicator of Cattle Herd Health Status? Approaches to Setting Reference Values

Andrzej Milczak [1], Beata Abramowicz [1,*], Marcin Szczepanik [1], Jacek Madany [1], Karolina Wrześniewska [1], Krzysztof Buczek [2], Marta Staniec [2], Paweł Żółkiewski [3] and Łukasz Kurek [1]

[1] Department and Clinic of Animal Internal Diseases, Faculty of Veterinary Medicine, University of Life Sciences in Lublin, Głęboka 30, 20-612 Lublin, Poland; andrzej.milczak@up.lublin.pl (A.M.); marcin.szczepanik@up.lublin.pl (M.S.); jacek.madany@up.lublin.pl (J.M.); karolina.wrzesniewska@up.lublin.pl (K.W.); lukasz.kurek@up.lubin.pl (Ł.K.)

[2] Department of Epizootiology and Clinic of Infectious Diseases, Faculty of Veterinary Medicine, University of Life Sciences in Lublin, Głęboka 30, 20-612 Lublin, Poland; krzysztof.buczek@up.lublin.pl (K.B.); marta.staniec@up.lublin.pl (M.S.)

[3] Department of Cattle Breeding and Genetic Resources Conservation, Faculty of Animal Sciences and Bioeconomy, University of Life Sciences in Lublin, Akademicka 13, 20-950 Lublin, Poland; pawel.zolkiewski@up.lublin.pl

[*] Correspondence: beata.abramowicz@up.lublin.pl

**Abstract:** Fibrinogen is used in the diagnosis of inflammation as an acute phase protein. The research objective set by the authors of this study was to assess the applicability of fibrinogen concentration measurement in the blood plasma of the peak of lactating dairy cows through the evaluation of the reference values by using the functional PT-derived (Prothrombin Time-derived) method. Materials and methods: The study was carried out on 259 HF (Holstein-Friesian) and white-backed cows. The animals were clinically healthy. Fibrinogen concentration was determined by automated PT-derived method. Fibrinogen concentrations were calculated as the mean of duplicate samples. Samples with differences between duplicate results greater than 5% were rejected. Results: In the group of HF cows, the average fibrinogen concentration was $11.75 \pm 4.80$ g/L. In white-backed cows, it was $9.53 \pm 4.79$ g/L. At total of 76.01% of the results of the fibrinogen concentration in HF cows and 82.05% of the results obtained in the group of white-backed cows were within the $\pm 1$ SD (Standard Deviation) range. Conclusions: Based on our own research, the PT-derived method may be applied in order to determine the concentration of fibrinogen in cattle herds in animal health monitoring studies. An individual laboratory should focus more on verifying reference intervals established elsewhere.

**Keywords:** fibrinogen; dairy cows; reference norms

## 1. Introduction

In 2022, the Organization for Economic Cooperation and Development (OECD) and the Food and Agriculture Organization of the United Nations (FAO) projected that global milk production would increase by 1.8% over the next decade [1]. Considering the reduction in the cattle population which has been observed in recent years and also feed price inflation, it is expected that achieving this goal will only become possible by increasing the productivity of dairy cows [1,2]. About 95% of the dairy cattle population in the European Union are HF cows with a high genetic potential, which is, however, not used to full advantage [3]. In 2020, the average annual milk yield per cow was over 7000 L, while the milk yield in individual HF breed cows may even reach 9000–11,000 L per year [2,3]. This situation is determined by a variety of factors including those that are health related. In the late 1980s, the foundation was laid for the Veterinary Herd Health Management (VHHM) programmes, which may be described as an integrated approach to all issues related to dairy farm management based on the collection of herd data. These activities were aimed at a more rapid evaluation

of the problems which may affect milk production [4–7]. Unfortunately, there have been some drawbacks with the adoption of this concept, including poor communication between the veterinarian and the farmer and a lack of detailed clarification concerning which herd data should be analysed [5–7]. Therefore, it is now also proposed to introduce inexpensive and simple point-of-care tests which would provide more information about the health of animals [5,6,8].

In the diagnosis of human and animal diseases, the determination of the concentration of acute phase proteins is a long-established method [9–12]. One such protein is fibrinogen, the concentration of which may increase from 2 to 10 times upon inflammation and infection [9]. Measurement of plasma fibrinogen levels could be easier and, therefore, more useful in everyday practice than those of other acute phase proteins. Fibrinogen is a major soluble plasma glycoprotein that plays a key role in haemostasis. The fibrinogen molecule consists of pairs of three polypeptide chains ($\alpha$, $\beta$, and $\gamma$) connected by disulphide bridges. This glycoprotein is synthesized in hepatocytes, this is why its concentration decreases with the onset of some liver diseases [12]. Fibrinogen also participates in the response to infection or injury by acting as a positive acute phase protein [9,10,13]. This is the reason for the measurement of fibrinogen levels being used as an indicator of pathological conditions. At present, a variety of methods are used to determine the concentration of fibrinogen in blood plasma [11,14–23]. Some of them are cheap and simple enough to even be used in field conditions [11,22,23]. It should be noted that different methods may produce different results. The physiological period and the age of the animals, as well as the location where the tests are performed, are also important factors [23,24]. Therefore, a possible solution to this problem may be to set the local standards for a specific region based on the use of a carefully selected method.

The research objective set by the authors of this study was to assess the applicability of fibrinogen concentration measurement in the blood plasma of the peak of lactating dairy cows through the evaluation of the reference values by using the functional PT-derived method.

## 2. Materials and Methods

### 2.1. The Research Sample

The study was carried out on 259 HF and white-backed cows, aged 3 to 6 years, with an average BCS (Body Condition Score) of about 3.5/5 points, at the same stage of lactation. The animals came from four indoor system farms in the Lublin region. The cows were divided into 2 groups based on their breed. The first group consisted of 222 HF breed animals with milk production of 9000–10,000 kg of milk per lactation. Animal nutrition in both farms was based on the TMR (Total Mixed Ration) method. The ration included: maize silage, ground and whole maize grain, haylage, grass silage, hay, straw, granulates, soybean meal, on-farm grown cereals, feed additives with a protein content of 18–24%, premixes, and mineral-vitamin supplements. The feed ration was formulated according to milk yield, the current physiological period, and the age and body weight of the cows. The second group consisted of white-backed cows; the study included 37 animals from two farms. The milk production of these cows ranged from 4000 to 5000 kg of milk per lactation. The nutrition provided was based on its own fodder composed of hay, straw, haylage, grass silage, maize silage, beet pulp, beet leaves silage, mineral and vitamin supplements, and salt licks. The owners of the tested animals did not report any health problems in the cows. No pathological changes were found upon clinical examination. The animals were under constant veterinary care. Hoof correction in the herds was performed twice a year and prophylactic deworming (spot-on delivery method, eprinomectin) was also carried out. Animal blood testing was performed as a part of the annual herd monitoring process. The perinatal period in the examined cows was normal and healthy calves were born. Animals with severe parturition and metabolic disorders in the postpartum period were excluded from the study, as well as cows in which parenchymal organ dysfunction was clearly detected in the period before the study. Moreover, excluded were those which had

not been prophylactically dewormed within the last year. Cows with health problems in the form of severe diarrhoea the presence of blood in the faeces and inflammation detected by clinical examination were also removed from the study group.

The minimum sample size was determined so that the maximum error would be kept as small as possible. The number of cows over 2 years old in Poland was over two million, (2,207,700) herds, which was taken as the basis for the calculations performed [25]. The population of white-backed cows was estimated at about 500 head [26]. The following formula was used for the estimation of the minimum sample size:

$$N_{min} = \frac{N_p\left(z^2 f(1-f)\right)}{N_p e2 + z^2 f(1-f)}$$

where

$N_{min}$—minimum sample size,

$N_P$—the size of the population from which the sample is taken,

$z$—assumed confidence level for the normal distribution and confidence interval of 0.95–1.96,

$f$—fraction size (assumed at 0.5),

$e$—assumed maximum error, expressed as a fractional number.

Due to the lack of certainty as to whether the results obtained for the studied parameter follow a normal distribution, they were also compared with the result of the minimum sample size, as given by the Slovin formula:

$$N_{min} = \frac{N_p}{1 + N_p e^2}.$$

### 2.2. Determination of Fibrinogen Concentration

The fibrinogen concentration was determined with the application of the PT-derived method using thromboplastin. An automatic optical coagulometer Bioksel 6000 (Bio-Ksel, Grudziądz, Poland) and Bio-Ksel reagents were used in this study. All of the analyses were performed on fresh citrated platelet-poor plasma, separated from citrated blood within 1 h of collection. Blood centrifugation was carried out at room temperature at 4500 g. A lyophilizate of standard human plasma was used to calibrate the coagulometer (Bio-Ksel, Poland).

Coagulometer was checked using as control a fresh pooled normal bovine platelet-poor plasma that had been assigned a fibrinogen level using von Clauss reference method. Fibrinogen concentration in both control and tested material were calculated as the mean of duplicate samples. Samples with differences between duplicate results greater than 5% were rejected.

### 2.3. Statistical Analysis

Statistical calculations were performed using Excel 2013 and the Statistic Kingdom online statistical calculator (https://www.statskingdom.com/index.html, accessed on 1 April 2023) [27]. The normality of the distribution was checked using the Shapiro–Wilk test. The differences between the studied populations were tested using the Whitney–Mann U test.

### 3. Results

The results are presented in Figure 1. In the group of HF cows, fibrinogen concentrations ranged from 3.54 g/L to 28.4 g/L, with an average result of 11.75 ± 4.80 g/L. The results obtained for the white-backed cows were similarly distributed. Reaching an average of 9.53 ± 4.79 g/L, the values of the examined parameter were in the range of 4.26–28.4 g/L. The medians were 10.44 g/L for HF cows and 8.44 g/L for white-backed cows. A total of 95% of the fibrinogen concentration results were in the range of 4.51–25.1 g/L for HF cows and 4.46–21.74 g/L for white-backed cows. In the group of HF cows, for 13 individuals, the

results obtained were higher than twice the standard deviation above the mean, and in the group of white-backed cows. A total of 76.01% of the results of the fibrinogen concentration in HF cows and 82.05% of the results obtained in the group of white-backed cows were within the ±1 SD range. These ranges were not statistically different at $\alpha$ = 0.05. Taking into consideration the sizes of the animal groups in this study, it may be reasonably assumed that in the case of the group of HF cows, the maximum error of the obtained results was less than 10%, and in the group of white-backed cows, the maximum estimation error did not exceed 15% at a confidence level set at 0.95. The series of results for the group of HF cows deviated from the normal distribution, and as the Shapiro–Wilk test showed, the data followed an asymmetric, right-sided, and slightly mesokurtic distribution. In the group of white-backed cows, the Shapiro–Wilk test confirmed the normality of the distribution ($p$ = 0.241 at $\alpha$ = 0.05). Due to the lack of a normal distribution of results in one of the groups, the Whitney–Mann test was used to verify the $H_o$ hypothesis about the insignificance of differences between the medians of the examined variable. At a significance level of $\alpha$ = 0.05, no statistical difference was found between the two study groups ($p$ = 0.1165).

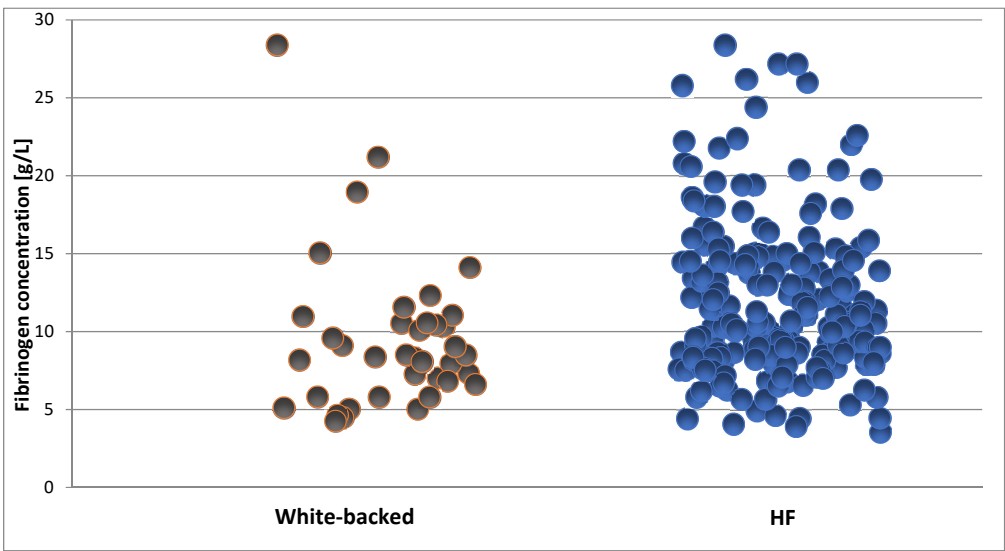

**Figure 1.** Fibrinogen concentration in blood of cows.

The time required for the automated measurement of 20 samples was shorter than 40 min (about 2 min/sample). True cost of analysis per sample was below 1€.

## 4. Discussion

Plasma fibrinogen concentration is dependent on many physiological factors and also on the test method used. It was most surprising for the authors to obtain a large dispersion of concentration values for this protein in these fairly homogeneous groups of animals and with the use of the same method each time. In 1970, McSherry et al. [12] established reference values for the fibrinogen concentration of different production groups of cattle. In the case of heifers and non-pregnant cows, these values were in the range of 3.8–11.2 g/L (6.6 ± 1.72 g/L) and, in the case of pregnant cows, the upper range of the norm established by the aforementioned authors was 13.5 g/L. In their studies, they determined the content of clotting protein in plasma using the biuret method. However, this method is not suitable for routine applications due to its laborious nature [23]. In more recent studies by various authors, the reference values were found to range from 3.0 to 7.0 g/L [9], and even lower [22]. Kritsepi-Konstantinou et al. [21] examined the fibrinogen concentration in 254 healthy HF cows, obtaining values in the range of 3.32–7.27 g/L. The method of thermal precipitation that they used for the measurement, although simple and convenient, can produce values that are up to 30% higher than those of the chronometric methods [28].

Comparing the results obtained using different methods in the case of fibrinogen concentration, measurement is not viable. In population studies, one method that is reproducible should be used. The method used by the authors of this paper, despite some limitations, is both reproducible and widely accepted. An additional advantage of the PT-derived methods is the possibility of automation, which is of great importance when examining a large number of animals. Those results obtained by the authors, which deviate from the average to a large extent, require further research. Clinical and subclinical forms of various diseases may cause a significant increase in fibrinogen levels [9,10,12,13,17,18,29]. These conditions are often not detected using routine clinical examination methods and the recommended screening tests. Heuwieser et al. [30] also paid attention to the effect of pregnancy on fibrinogen levels, while Corbiere [31] noted a gradual increase in fibrinogen concentration with age. In the present study, the importance of the latter factor seems unlikely due to the fairly uniform age of the herds studied, and the fact that a noticeable increase in fibrinogen concentration may be detected three days before and one day after parturition [30].

An overly wide range of accepted reference values makes the examined parameter clinically useless. There may be many reasons for the difficulties involved in defining a valid reference range. Choosing the right population is not a simple task as the standard definition of "good health" is particularly difficult to establish [32,33]. The results obtained even for the same individual are subject to variability [32]. There are also different approaches to the determination of the range of normal values. It is usually proposed for the results to be within $\pm 2$ SD from the mean, although some authors have narrowed this down to $\pm 1$ SD [32]. In 2016, the Panel of Experts on Reference Values of the International Federation of Clinical Chemistry and Laboratory Medicine (IFCC) published, under the auspices of the IFCC and the Clinical and Laboratory Standards Institute (CLSI), the updated guidelines concerning methods for the determination of reference limits [33]. The authors of these guidelines formulated detailed algorithms to solve the problems associated with determining the range of normal values. Based on the information above, in our own research, it was established that the most precise standard for HF cows is 6.95–16.55 g/L, and for white-backed cows it was 4.74–14.32 g/L using the $\pm 1$ SD range. Our research shows that although procedures for establish reference ranges in the literature are clear in theory, gaps remain for the implementation of these procedures in routine veterinary practice. Dairy herds also continue to pose additional challenges in respect of acquiring and verifying reference ranges for fibrinogen concentration.

## 5. Conclusions

Early identification of cows with a higher risk of health problems is crucial for dairy farms to prevent and ameliorate the negative economic effects of these disorders in a timely manner. Laboratory testing panels for cattle currently lack acute phase proteins to indicate the health status of these animals. The laboratory tests available for assessing inflammatory disease comprise measurements of acute phase protein response such as serum amyloid A, haptoglobin, and alpha 1-acid glycoprotein though, their use in everyday practice is not profitable from the economic point of view due to high costs. The PT-derived fibrinogen test is simple and inexpensive and can be widely used with automated methods. Fibrinogen can facilitate the diagnosis of animals with subclinical and atypical abnormalities, especially in their early stages. In this case, however, it is necessary to determine the correct range of values of this parameter to which the obtained results can be related. The authors of this article pointed to the wider than mentioned in the literature reference ranges of fibrinogen concentration in healthy dairy cows and so far, lack of explain this discrepancy. This study is part of research aimed at monitoring, predicting, and early detection of health disorders in cows in commercial herds.

**Author Contributions:** Conceptualization, A.M., B.A. and Ł.K.; Methodology, A.M. and B.A.; Software, A.M.; Validation, A.M., K.W. and K.B.; Formal Analysis, A.M., B.A., M.S. (Marcin Szczepanik), Ł.K., K.W. and J.M.; Investigation, B.A., Ł.K., K.B., M.S. (Marta Staniec), P.Ż.,; Data Curation, A.M., B.A., Ł.K., M.S. (Marta Staniec), K.B. and P.Ż.; Writing—Original Draft Preparation, A.M.; Writing—Review & Editing, B.A., Ł.K., K.W. and M.S. (Marcin Szczepanik); Visualization, A.M., B.A. and P.Ż.; Supervision, J.M.; Project Administration, B.A.; All authors have read and agreed to the published version of the manuscript.

**Funding:** This work was supported by the University of Life Sciences in Lublin, project No. WKW/S/31/2023.

**Institutional Review Board Statement:** Not applicable. Animal blood testing was performed as a part of the annual herd monitoring process.

**Data Availability Statement:** The data reported in this study are contained within the article.

**Conflicts of Interest:** The authors declare no conflict of interest.

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
