# Peer review of "Could Fibrinogen Concentration Be a Useful Indicator of Cattle Herd Health Status? Approaches to Setting Reference Values"

_agriculture, doi:10.3390/agriculture13061224_

Round 1

Reviewer 1 Report

My main point lies in the "benefit" of the research conducted. Fibrinogen values have been monitored in two populations of dairy cows, but it is not clear why this method is useful.

Cheaper; faster; more accurate than other methods. The study has not been tested in inflammatory processes of any nature (viral, bacterial, parasitic or other).

All this gives me reason not to accept the manuscript as it is because it is simply worthless.

Author Response

Dear Editor,

We appreciate the time and effort that you and the reviewers dedicated to providing feedback on ourmanuscript and are grateful for the insightful comments on and valuable improvements to our paper. We have incorporated most of the suggestions made by the reviewers. Those changes are highlighted in red within the manuscript.

Reviewer 1st:

While we appreciate the reviewer’s feedback, we respectfully disagree with his final conclusion. We have the impression that the reviewer read our work slightly cursory. Perhaps the initial title and abstract was misleading. Of course, we have restructured this parts for better understanding. As indicated in the text, we are referring strictly to healthy animals. Clinically sick cows were excluded. Animals with subclinical diseases were also excluded from the study using available laboratory tests. High levels of fibrinogen in potentially healthy cows are puzzling, aren't they? Moreover, the conclusions did not state that the method is unconditionally useful.

Reviewer 2 Report

The communication entitled "Evaluation of the usefulness of fibrinogen concentration measurement in dairy cows using the PT-derived method" is interesting according to my opinion, while the manuscript is less of line numbers, leading to some difficulties to accurately point out the source of errors or comments for peer review. Some issues need to be better argued further, and my questions are as follows.

1.     The title of the manuscript: “Evaluation of the usefulness of…”. In fact, the authors did little work on the “usefulness” but only “evaluation of the reference values”. So, the title could be further optimized.

2.     Full names of some abbreviations in the manuscript, such as PT, HF, BCS, should be given upon their first appearance. Similarly, the abbreviation of “standard deviation”, SD, should be given upon its first appearance.

3.     Please note the type size of the sentence “In the late 1980s,…” in the Introduction section.

4.     The last sentence in Abstract, “for the herd of region”?

5.     “g/l” should be “g/L”, like it in the figure.

6.     The figure legend was missed.

7.     I suggest to add the Table data for the manuscript, at least a supplemental table.

8.     Please pay attention to some white spaces in the manuscript, [27], [31]…

9.     It seems that we are still not clear on the usefulness of fibrinogen assessment in dairy cows.

10.   Could authors provide the evidence for the good reproducibility of the PT method?

11.   “Conclusions” could be further simplified.

Reviewer 3 Report

-Nice work but insufficient and needed substantial improvement for clarity and correctness. The novelty of the study is poorly visible. The methodology is poorly presented. Straightforward conclusion is unclear and hardly visible.

-What was the main basis for choosing the fibrinogen (very old version and, in terms of quantity it is widely ranged and very general and. Please address in the methods how did you diagnose and differentiate the diseased cows with such measurements? What is the cut-off concertation for diseased high yielding dairy cows? Please elaborate?  

-Some parts of the text are unclear and ambiguous; the authors should revise and improve the text for the clarity throughout.

-Abstract is not informative and very insufficient and unclear to get the point.

-The abbreviations in the abstract should be defined.

- the cows in the study group was fairly homogeneous groups?  Are you sure? way big variations? The parameters should be clinically and para-clinically useful and applicable.

-the concentration of targeted parameter studied here is hugely in big range! This is a big issues to have a cut-off for healthy versus diseased cows! Please elaborate this big issues of the study plan and reader might want to see what is the point of measuring such a high-ranged value parameter!

-...The number of cows over 2 years old, amounting to 2,207.7 thousand, ... unclear! what does it mean?

-...4,000 to 5,000 kg of milk....? per lactation?

Please see above ... in general it should be improved substantially.
